# Embedding-Assisted Entity Resolution for Knowledge Graphs*

Daniel Obraczka[0000−0002−0366−9872], Jonathan Schuchart[0000−0003−3507−8150], and Erhard Rahm[0000−0002−2665−1114]

Leipzig University, Germany
{obraczka,schuchart,rahm}@informatik.uni-leipzig.de

**Abstract.** Entity Resolution (ER) is a main task for integrating different knowledge graphs in order to identify entities referring to the same real-world object. A promising approach is the use of graph embeddings for ER in order to determine the similarity of entities based on the similarity of their graph neighborhood. Previous work has shown that the use of graph embeddings alone is not sufficient to achieve high ER quality. We therefore propose a more comprehensive ER approach for knowledge graphs called EAGER (*E*mbedding-*A*ssisted Knowledge *G*raph *E*ntity *R*esolution) to flexibly utilize both the similarity of graph embeddings and attribute values within a supervised machine learning approach and that can perform ER for multiple entity types at the same time. Furthermore, we comprehensively evaluate our approach on 19 benchmark datasets with differently sized and structured knowledge graphs and use hypothesis tests to ensure statistical significance of our results. We also compare our approach with state-of-the-art ER solutions, where EAGER yields competitive results for shallow knowledge graphs but much better results for deeper knowledge graphs.

**Keywords:** Knowledge Graph · Knowledge Graph Embedding · Entity Resolution

## 1 Introduction

Knowledge Graphs (KGs) store real-world facts in machine-readable form. This is done by making statements about entities in triple form $(entity, property, value)$. For example the triple (`Get_Out, director, Jordan_Peele`) tells us that the director of the movie "Get Out" is "Jordan Peele". Such structured information can be used for a variety of tasks such as recommender systems, question answering and semantic search. For many KG usage forms including question answering it is beneficial to integrate KGs from different sources. An integral part of this

---

* This work was supported by the German Federal Ministry of Education and Research (BMBF, 01/S18026A-F) by funding the competence center for Big Data and AI "ScaDS.AI Dresden/Leipzig". Some computations have been done with resources of Leipzig University Computing Center.

integration is entity resolution (ER), where the goal is to find entities which refer to the same real-world object.

Existing ER systems mostly focus on matching entities of one specific entity type (e.g. publication, movie, customer etc.) and assume matched schemata for this entity type. This proves challenging when trying to use these systems for ER in KGs typically consisting of many entity types with heterogeneous attribute (property) sets. See Figure 1 for an example showcasing the many different challenges of this task, such as heterogeneous date representations ("1979-02-21" vs. "21 Febuary 1979"), differing URIs ("dbr:Jordan Peele" vs. "wd: Q3371986"), schemata ("dbo:birthdate" vs. "wdt:P569") and overall information contained.

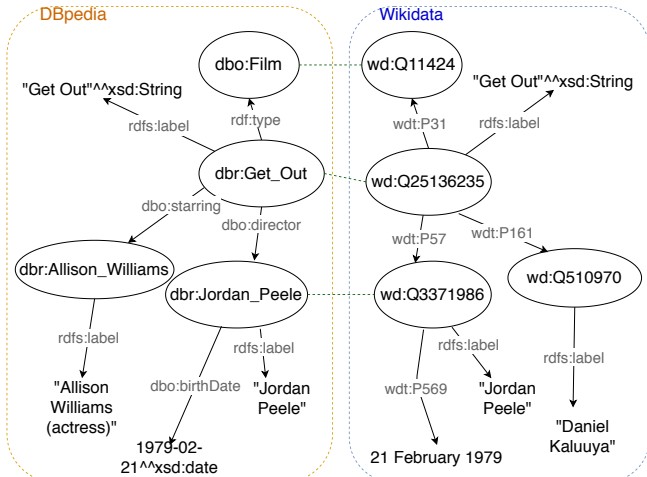

Fig. 1: Subgraphs of DBpedia and Wikidata. Green dashed lines show entities that should be matched. Some URIs are shortened for brevity.

We observe there are entities of different types (film, director, actor) and different attributes with heterogeneous value representations (e.g., birth date values "1979-02-21" in DBpedia and "21 Febuary 1979" in Wikidata for two matching director entities). Moreover, we see that matching entities such as the movie "Get Out" have different URIs and differently named edges referring to properties and related entities, e.g. `rdf:type` vs. `wdt:P31`. These aspects make a traditional schema (property) matching as a means to simplify ER very challenging so that entity resolution for KGs should ideally not depend on it. Given that URIs and property names may not show any similarity it becomes apparent that the graph structure and related entities should be utilized in the ER process, e.g., to consider the movie label and director to match movies.

A promising way to achieve this in a generic manner, applicable to virtually any entity type, is the use of graph embeddings. By encoding the entities of the KGs into a low-dimensional space such approaches alleviate the obstacles

posed by the aforementioned KG heterogeneities. Capturing the topological and semantic relatedness of entities in a geometric embedding space enables the use of these embeddings as inputs for machine learning (ML) algorithms. The performance of graph embedding approaches for ER has been recently studied by Sun et. al [28]. However, as they point out, most approaches focus on refining the embedding process, while ER mostly consists of finding the nearest neighbors in the embedding space. Hence, the use of graph embeddings has to be tailored to the ER task for good effectiveness. We build on the findings of  [28] and investigate the usefulness of learned graph embeddings as input for ML classifiers for entity resolution. While there are different settings for KG integration, such as enhancing a given KG or KG fusion, we focus here on the simple ER setting, i.e., finding matching entities in two data sources. The resulting match mappings can then be used for applications such as question answering or as input for KG fusion.

In this paper, we propose and comprehensively evaluate the first (to our knowledge) *graph embedding supported* ER system named **EAGER**: **E**mbedding **A**ssisted Knowledge **G**raph **E**ntity **R**esolution. It uses both knowledge graph embeddings and attribute similarities as inputs for an ML classifier for generic entity resolution with several entity types. EAGER utilizes different kinds of graph embeddings, specifically the ones that performed best in [28], as well as different ML classifiers. We comprehensively evaluate the match effectiveness of EAGER with using graph embeddings and attribute similarities either alone or in combination for 23 datasets of varying size and structure. Some of these ER tasks with multiple entity types from the movie domain have been newly developed for this work. The evaluation also includes a comparison of EAGER with state-of the-art ER approaches, namely Magellan [16] and DeepMatcher [19]. All our results are analyzed using hypothesis tests to ensure statistical significance of our findings.

We first discuss related work followed by an overview of EAGER. Section 4 describes the used datasets including the new benchmarks from the movie domain. Our evaluation is presented in Section 5 and we conclude in Section 6.

## 2   Related Work

Entity resolution has attracted a significant amount of research, sometimes under different names such as record linkage [8,10], link discovery [29,26] or deduplication [25]. We focus on the discussion of the most related ER approaches. and refer to surveys and books such as  [11,20,6] for a more thorough overview. Traditional ER approaches rely on learning distance- or similarity-based measures and then use a threshold or classifier to decide about whether two entities are the same. These classifiers can be unsupervised [22,23], supervised [26,14] or employ active learning [25,21]. For example the Magellan Framework [16] provides supervised ML classifiers and provides extensive guides for the entire ER process. Recently, deep learning has seen some success in certain settings. DeepER [9] and DeepMatcher [19] provide a variety of different architectures and among

other aspects, such as attribute similarities, use word embeddings as inputs for these networks. Both frameworks have shown that especially for unstructured textual data deep learning can outperform existing frameworks.

Collective ER approaches try to overcome the limitations of the more conventional attribute-based methods. This paradigm uses the relationships between entities as additional information and in some cases even considers previous matching decisions in the neighborhood. Bhattacharya and Getoor [3] show that using the neighborhood of potential match candidates in addition to attribute-based similarity is especially useful for data with many ambiguous entities. SiGMa [17] uses an iterative graph propagation algorithm relying on relationship information as well as attribute-based similarity between graphs to integrate large-scale knowledge bases. Pershina et al. [24] propagate similarities using Personalized PageRank and are able to align industrial size knowledge graphs. Zhu et al. [32] reformulate entity resolution as multi-type graph summarization problem and use attribute-based similarity as well as structural similarity, i.e. connectivity patterns in the graph.

More recently the use of graph embeddings has been shown promising for the integration of KGs. An overview of currently relevant approaches that solely rely on embedding techniques can be found in [28]., some of these techniques have been used in this work and will be discussed in more detail in Section 3.3. Knowledge graph embedding (KGE) models typically aim to capture the relationship structure of each entity in latent vector representations in order to be used for further downstream applications. For an overview of current knowledge graph embedding approaches we refer the reader to a recent survey from Ali et al. [1].

EAGER aims to combine the two generally separate ER approaches of graph embedding techniques and traditional attribute- based methods for the integration of KGs with multiple entity types without relying on additional schema matching or any structural assumptions about the entities. Our extensive evaluation for a large spectrum of KGs demonstrates the viability of the proposed approach.

## 3   Overview of EAGER

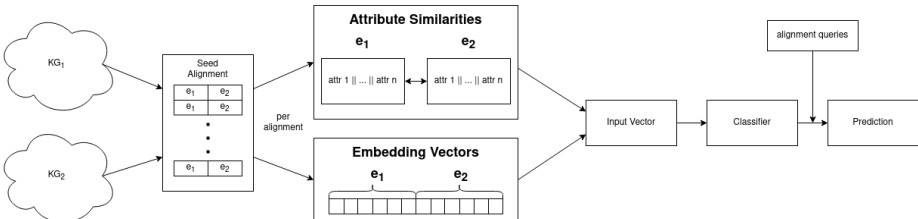

Fig. 2: Schematic summary of EAGER

In this section we present an overview of the EAGER approach for ER in knowledge graphs and the specific approaches and configurations we will evaluate.[1] We start with a formal definition of the ER problem and an overview of the EAGER workflow. Subsequently we explain how we create the input vector for our ML classifiers and conclude with a discussion of the prediction step.

### 3.1   Problem statement

KGs are constructed by triples in the form of $(entity, property, value)$, where *property* can be either a attribute property or a relationship and *value* a literal or another entity, respectively. Therefore, a KG is a tuple $\mathcal{KG} = (\mathcal{E}, \mathcal{R}, \mathcal{A}, \mathcal{L}, \mathcal{T})$, where $\mathcal{E}$ is the set of entities, $\mathcal{A}$ the set of attribute properties, $\mathcal{R}$ the set of relationship properties, $\mathcal{L}$ the set of literals and $\mathcal{T}$ is the set of triples. We distinguish attribute triples $\mathcal{T}_A$ and relationship triples $\mathcal{T}_R$, where $\mathcal{T}_A : \mathcal{E} \times \mathcal{A} \times \mathcal{L}$ are triples connecting entities and literals, e.g. (`dbr:Jordan_Peele, dbo: birthDate, "1979-02-21"`) and $\mathcal{T}_R : \mathcal{E} \times \mathcal{R} \times \mathcal{E}$ connect entities, e.g. (`dbr: Get_Out, dbo:director, dbr:Jordan_Peele`) as seen in Figure 1. Our goal is to find a mapping between entities of two KGs. More formally, we aim to find $\mathcal{M} = \{(e_1, e_2) \in \mathcal{E}_1 \times \mathcal{E}_2 | e_1 \equiv e_2\}$, where $\equiv$ refers to the equivalence relation. Furthermore, we assume we are provided with a subset of the mapped entities $\mathcal{M}_T \subseteq \mathcal{M}$ as training data, which is also sometimes referred to as seed alignment in the literature.

### 3.2   Overview

The remaining chapter is dedicated to illustrate how our approach tackles entity resolution in heterogeneous KGs. A schematical overview can be found in Figure 2. Given two KGs $\mathcal{KG}_1, \mathcal{KG}_2$ and a set of initial matches $\mathcal{M}_T$ we create a feature vector for each match $(e_1, e_2) \in \mathcal{M}_T$ to train a machine learning classifier. Additionally to the positive matches provided in $\mathcal{M}_T$ we sample negative examples by sampling random pairs $(e_1, e_2) \notin \mathcal{M}_T$ to create a balanced set of positive and negative examples. After the training step the classifier then acts as an oracle to answer specific alignment queries, i.e. entity pairs, in order to make a prediction. In the following we present our approach in more detail.

### 3.3   Input Vector Creation

Since schemata across different KGs may differ wildly, creating a schema matching before ER in heterogeneous KGs is difficult and can introduce additional sources for error. Keeping the focus on the matching process, we chose to concatenate all attribute values of each entity into a single string and used 3 similarity measures for comparisons: Levenshtein, Generalized Jaccard with an Alphanumeric Tokenizer, which returns the longest strings of alphanumeric characters,

---

[1] The code for EAGER and our experiments can be found in `https://github.com/ jonathanschuchart/eager`

and Trigrams with the Dice coefficient. The second part of the input vector consists of KGEs. Given that the focus of this study lies not on the creation of embeddings itself, our approach can take any entity embeddings that are embedded in the same space. Since most KG embedding frameworks are not specialized for ER, we use OpenEA[2] which was developed by Sun et al. for their 2020 benchmark study[28]. It offers a variety of embedding approaches and embeds entities into the same space. Specifically, we chose three of the best approaches of said study, namely BootEA, MultiKE and RDGCN:

**BootEA** Sun et al. in 2018 [27] based their approach on the TransE model and combined it with elaborate bootstrapping and negative sampling techniques to improve performance. TransE aims to find an embedding function $\phi$ that minimizes $||\phi(e_h) + \phi(r) - \phi(e_t)||$ for any $(e_h, r, e_t) \in \mathcal{T}_R$. Bootstrapping is done by additionally sampling likely matching entities (resampled every few epochs based on the current model) in order to increase the effective seed alignment size. Additionally, negative relationship tuples are sampled and resampled every few epochs based on the current model in order to improve the distinction between otherwise similar entities. Since TransE is an unsupervised model, Sun et al. proposed a new objective function which incorporates both the original objective function of TransE and the likelihood of two entities from different KGs matching.

**MultiKE** In order to also incorporate more than just relational information, Zhang et al. [31] proposed a flexible model which combines different views on each entity. Here, the name attribute, relations and all remaining attributes are embedded separately, using pre-trained word2vec word embeddings [18] for names and a variation on TransE for relations. Attribute embeddings are obtained by training a convolutional neural network taking the attribute and attribute value as input. All three embedding vectors are then combined into a single unified embedding space. In this approach the two knowledge graphs are treated as one combined graph where entities from the seed alignment are treated as equal.

**RDGCN** Different to the aforementioned approaches, Wu et al. [30] proposed a new technique using two constructed conventional graphs and the GCN model by Kipf and Welling with highways. Instead of learning embeddings for entities and relations within one graph, RDGCN constructs a primary entity graph and a dual relationship graph in order to alternate the optimization process between the two. That way, the relationship representations from the dual graph are used to optimize the entity representations from the primal graph and vice versa by applying a graph attention mechanism. As the actual neigborhood information of each entity is not fully exploited in this case, Wu et al. showed that feeding the resulting entity representations into a GCN can help significantly improve the overall embedding quality.

---

[2] https://github.com/nju-websoft/OpenEA

### 3.4    Combinations

As the aim of our study is to investigate to what degree combining entity embeddings with attribute similarities is superior to using either on their own, we present three different variants of our approach, that only differ in the construction of their input vector: $EAGER_E$ contains solely the embeddings, $EAGER_A$ consists exclusively of the attribute similarities and finally $EAGER_{A\|E}$ which is a concatenation of entity embeddings and attribute similarities. The respective input vector is given to a classifier along with the seed alignment. In the evaluation we achieved the best results using either a Multilayer Perceptron (MLP) [13] or Random Forest (RF) [4], but any classifier can be used.

### 3.5    Prediction

The trained classifier is presented with alignment queries, i.e. pairs of entities that it will have to classify as match or non-match. Choosing these pairs is a non-trivial question since exploring all possible pairs would lead to a quadratic number of alignment queries relative to the KG size, which is not scalable to large datasets. Traditionally, blocking strategies are used to reduce the number of pairs by a linear factor. Due to the heterogeneous nature of KGs new strategies for this problem have to be found. An alternative could be to use the embeddings to find a number of nearest neighbors, which is a scalable solution since the triangle inequality in metric spaces can be exploited to reduce the number of comparisons for the neighborhood search. Finding a good solution for this problem is however out of scope for our study and in the experiments we therefore use the test data to create prediction pairs, sampling negative examples randomly as done in the training step. More on our experimental setup can be found in Section 5.1.

## 4    Datasets

To evaluate our approach we use multiple datasets that can generally be put into two categories: *rich* and *shallow* graph datasets. The rich graph datasets were presented in [28] and consist of samples from DBpedia (D), Wikidata (W) and Yago (Y). Given their origin in web-scale KGs they offer a wide range of relationships as well as entity types. The linking tasks include KG samples of different density, size, as well as covering cross-lingual settings (EN-DE & EN-FR).

   To investigate how the interplay of attribute similarities and graph embeddings fares in settings with less dense KGs we created a new benchmark dataset with multiple entity types. These KGs are taken from the movie domain, where the gold standard was hand-labeled for the five entity types `Person, Movie, TvSeries, Episode, Company`. The movie datasets were created from three sources containing information about movies and tv series: IMDB[3], The-

---

[3] https://www.imdb.com/

MovieDB[4] and TheTVDB[5]. We make the movie datasets publicly available for future research at `https://github.com/ScaDS/MovieGraphBenchmark`. More details on each of the datasets can be found in Table 1 and Table 2 respectively.

Table 1: Shallow graph datasets statistics

| Datasets | KGs | $|\mathcal{R}|$ | $|\mathcal{A}|$ | $|\mathcal{T}_R|$ | $|\mathcal{T}_A|$ | $|\mathcal{E}|$ | $|\mathcal{M}|$ |
|---|---|---|---|---|---|---|---|
| imdb-tmdb | imdb | 3 | 13 | 17532 | 25723 | 5129 | 1978 |
| | tmdb | 4 | 493 | 27903 | 24695 | 6056 | |
| imdb-tvdb | imdb | 3 | 13 | 17532 | 25723 | 5129 | 2488 |
| | tvdb | 3 | 350 | 15455 | 21430 | 7810 | |
| tmdb-tvdb | tmdb | 4 | 493 | 27903 | 24695 | 6056 | 2483 |
| | tvdb | 3 | 350 | 15455 | 21430 | 7810 | |

Table 2: Rich graph datasets statistics, adapted from [28]

| | Datasets | KGs | V1 | | | | | V2 | | | | |
|---|---|---|---|---|---|---|---|---|---|---|---|---|
| | | | $|\mathcal{R}|$ | $|\mathcal{A}|$ | $|\mathcal{T}_R|$ | $|\mathcal{T}_A|$ | $|\mathcal{M}|$ | $|\mathcal{R}|$ | $|\mathcal{A}|$ | $|\mathcal{T}_R|$ | $|\mathcal{T}_A|$ | $|\mathcal{M}|$ |
| 15K | D-W | DB | 248 | 342 | 38,265 | 68,258 | 15,000 | 167 | 175 | 73,983 | 66,813 | 15,000 |
| | | WD | 169 | 649 | 42,746 | 138,246 | | 121 | 457 | 83,365 | 175,686 | |
| | D-Y | DB | 165 | 257 | 30,291 | 71,716 | 15,000 | 72 | 90 | 68,063 | 65,100 | 15,000 |
| | | YG | 28 | 35 | 26,638 | 132,114 | | 21 | 20 | 60,970 | 131,151 | |
| | EN-DE | EN | 215 | 286 | 47,676 | 83,755 | 15,000 | 169 | 171 | 84,867 | 81,988 | 15,000 |
| | | DE | 131 | 194 | 50,419 | 156,150 | | 96 | 116 | 92,632 | 186,335 | |
| | EN-FR | EN | 267 | 308 | 47,334 | 73,121 | 15,000 | 193 | 189 | 96,318 | 66,899 | 15,000 |
| | | FR | 210 | 404 | 40,864 | 67,167 | | 166 | 221 | 80,112 | 68,779 | |
| 100K | D-W | DB | 413 | 493 | 293,990 | 451,011 | 100,000 | 318 | 328 | 616,457 | 467,103 | 100,000 |
| | | WD | 261 | 874 | 251,708 | 687,860 | | 239 | 760 | 588,203 | 878,219 | |
| | D-Y | DB | 287 | 379 | 294,188 | 523,062 | 100,000 | 230 | 277 | 576,547 | 547,026 | 100,000 |
| | | YG | 32 | 38 | 400,518 | 749,787 | | 31 | 36 | 865,265 | 855,161 | |
| | EN-DE | EN | 381 | 451 | 335,359 | 552,750 | 100,000 | 323 | 326 | 622,588 | 560,247 | 100,000 |
| | | DE | 196 | 252 | 336,240 | 716,615 | | 170 | 189 | 629,395 | 793,710 | |
| | EN-FR | EN | 400 | 466 | 309,607 | 497,729 | 100,000 | 379 | 364 | 649,902 | 503,922 | 100,000 |
| | | FR | 300 | 519 | 258,285 | 426,672 | | 287 | 468 | 561,391 | 431,379 | |

## 5   Evaluation

We discuss our results on the presented datasets, starting with a description of the experiment setup, followed by the results on the shallow and rich graph

---

[4] `https://www.themoviedb.org/`

[5] `https://www.thetvdb.com/`

datasets, with a focus on investigating whether the use of attribute similarities in combination with knowledge graph embeddings is beneficial for the respective setting. Furthermore we compare our approach with state-of-the-art frameworks.

### 5.1   Setup

For the evaluation we use a 5-fold cross validation with a 7-2-1 split in accordance with [28]: For each dataset pair the set of reference entity matches is divided into 70% testing, 20% training and 10% validation. For each split we sample negative examples to create an equal share of positive and negative examples. The entire process is repeated 5 times to create 5 different folds.

For the OpenEA datasets the graph embeddings were computed using the hyperparameters given by the study of [28]. For all other datasets the *-15K parameter sets were used. For the classifiers, mostly scikit-learn's default parameters were used, though Random Forest Classifier was used with 500 estimators and MLP used two hidden layers of size 200 and 20. Furthermore, MLP was trained using the Adam [15] optimizer with $\alpha = 10^{-5}$.

### 5.2   Results

Table 3: Averaged F-measure on test set of rich graph datasets. The best value in a row is highlighted. For average rank the best 3 values of the compared ranks are highlighted

| Dataset | EAGER$_{A\|E}$ BootEA | | MultiKE | | RDGCN | | EAGER$_A$ | | EAGER$_E$ BootEA | | MultiKE | | RDGCN | |
|---|---|---|---|---|---|---|---|---|---|---|---|---|---|---|
| | MLP | RF | MLP | RF | MLP | RF | MLP | RF | MLP | RF | MLP | RF | MLP | RF |
| imdb-tmdb | 0.967 | 0.977 | 0.988 | 0.984 | 0.969 | 0.975 | 0.979 | 0.980 | 0.874 | 0.859 | 0.911 | 0.913 | 0.874 | 0.873 |
| imdb-tvdb | 0.938 | 0.960 | 0.973 | 0.967 | 0.940 | 0.953 | 0.965 | 0.960 | 0.821 | 0.786 | 0.873 | 0.844 | 0.807 | 0.792 |
| tmdb-tvdb | 0.973 | 0.977 | 0.983 | 0.981 | 0.966 | 0.977 | 0.980 | 0.978 | 0.874 | 0.844 | 0.871 | 0.877 | 0.857 | 0.831 |
| D-W(V1) (15K) | 0.775 | 0.668 | 0.881 | 0.858 | 0.805 | 0.842 | 0.827 | 0.828 | 0.764 | 0.678 | 0.853 | 0.871 | 0.718 | 0.707 |
| D-W(V2) | 0.934 | 0.841 | 0.945 | 0.918 | 0.897 | 0.890 | 0.868 | 0.870 | 0.938 | 0.847 | 0.939 | 0.942 | 0.808 | 0.796 |
| D-Y(V1) | 0.870 | 0.775 | 0.986 | 0.982 | 0.974 | 0.986 | 0.972 | 0.971 | 0.837 | 0.746 | 0.952 | 0.941 | 0.947 | 0.953 |
| D-Y(V2) | 0.983 | 0.908 | 0.995 | 0.993 | 0.977 | 0.991 | 0.978 | 0.978 | 0.975 | 0.888 | 0.973 | 0.971 | 0.947 | 0.960 |
| EN-DE(V1) | 0.923 | 0.852 | 0.986 | 0.984 | 0.966 | 0.976 | 0.947 | 0.945 | 0.891 | 0.798 | 0.957 | 0.950 | 0.937 | 0.955 |
| EN-DE(V2) | 0.970 | 0.918 | 0.992 | 0.990 | 0.968 | 0.978 | 0.956 | 0.955 | 0.946 | 0.875 | 0.961 | 0.958 | 0.934 | 0.956 |
| EN-FR(V1) | 0.868 | 0.736 | 0.978 | 0.973 | 0.950 | 0.963 | 0.922 | 0.920 | 0.806 | 0.709 | 0.952 | 0.942 | 0.907 | 0.935 |
| EN-FR(V2) | 0.965 | 0.876 | 0.991 | 0.989 | 0.963 | 0.977 | 0.937 | 0.936 | 0.942 | 0.875 | 0.977 | 0.978 | 0.921 | 0.948 |
| D-W(V1) (100K) | 0.873 | 0.850 | 0.887 | 0.862 | 0.768 | 0.774 | 0.810 | 0.811 | 0.868 | 0.820 | 0.850 | 0.871 | 0.645 | 0.556 |
| D-W(V2) | 0.962 | 0.927 | 0.951 | 0.923 | 0.756 | 0.792 | 0.845 | 0.844 | 0.959 | 0.916 | 0.917 | 0.957 | 0.610 | 0.609 |
| D-Y(V1) | 0.980 | 0.958 | 0.990 | 0.987 | 0.991 | 0.993 | 0.975 | 0.975 | 0.959 | 0.942 | 0.949 | 0.954 | 0.963 | 0.968 |
| D-Y(V2) | 0.993 | 0.965 | 0.995 | 0.990 | 0.983 | 0.989 | 0.976 | 0.975 | 0.979 | 0.958 | 0.953 | 0.978 | 0.921 | 0.968 |
| EN-DE(V1) | 0.943 | 0.907 | 0.989 | 0.982 | 0.954 | 0.961 | 0.944 | 0.943 | 0.901 | 0.859 | 0.956 | 0.947 | 0.872 | 0.891 |
| EN-DE(V2) | 0.965 | 0.933 | 0.993 | 0.988 | 0.926 | 0.932 | 0.943 | 0.941 | 0.934 | 0.890 | 0.970 | 0.969 | 0.779 | 0.847 |
| EN-FR(V1) | 0.925 | 0.867 | 0.981 | 0.969 | 0.947 | 0.938 | 0.920 | 0.919 | 0.866 | 0.819 | 0.948 | 0.943 | 0.866 | 0.894 |
| EN-FR(V2) | 0.968 | 0.899 | 0.989 | 0.979 | 0.897 | 0.901 | 0.925 | 0.923 | 0.925 | 0.877 | 0.959 | 0.968 | 0.742 | 0.806 |
| Avg Rank | 6.211 | 10.105 | 1.316 | 2.842 | 6.895 | 5.474 | 6.947 | 7.632 | 8.947 | 12.789 | 6.737 | 6.211 | 12.000 | 10.895 |

The results for all datasets are displayed in Table 3. In the bottom row we display the average rank of each combination of input variant, embedding

approach and classifier which is a number between 1 and 14 (since there are 14 possible combinations), where 1 would mean this combination achieves the best result for each dataset.

For the movie datasets we can see that $\text{EAGER}_{A\|E}$ with MultiKE performs best, especially with the MLP classifier. This suggests that even for datasets with relatively few relational information it can be beneficial to use knowledge graph embeddings. However, it seems very dependent on the way these embeddings are constructed in order for them to be useful with MultiKE being the only one of the three embedding approaches to explicitly incorporate attribute data. This also becomes apparent when comparing the results of $\text{EAGER}_E$ and $\text{EAGER}_A$ where MultiKE performs best for $\text{EAGER}_E$ but is still clearly outperformed by $\text{EAGER}_A$.

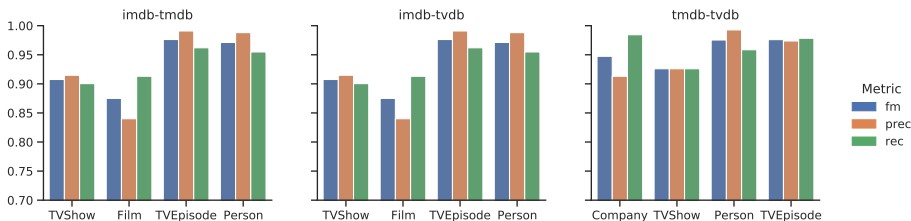

Fig. 3: Averaged F-measure, Precision and Recall per Type on Movie Datasets using $\text{EAGER}_{A\|E}$ with RF

Looking at the movie datasets in more detail as shown in Figure 3, we can see that there is a difference in performance depending on the entity type. In most cases, $\text{EAGER}_{A\|E}$ reaches an F-measure of over 90% for all entity types showing that the approach is generic and able to achieve good match quality for multiple heterogeneous entity types. Still there are some differences between the entity types. TVShows and Films generally perform worse than TVEpisodes and Persons with especially the precision for Film standing out negatively. This is especially pronounced in the IMDB-TMDB and IMDB-TVDB datasets. This might be attributed to different sets of attributes between those datasets, e.g. as IMDB does not contain full-length descriptions of films and tv shows whereas TMDB and TVDB do. Interestingly, Films/TVShows with very dissimilar titles due to different representations of non-English titles can be matched using the KGEs. For example the soviet drama "Defence Counsel Sedov" has the romanized title "Zashchitnik Sedov" in IMDB, while TMDB has either the translated "Defence Counsel Sedov" or the cyrillic "Защитник Седов". These entity pairs are correctly matched in the $\text{EAGER}_{A\|E}$ variant.

Looking at the rich datasets it is again evident that $\text{EAGER}_{A\|E}$ achieves the best results. Overall it can solve the diverse match tasks including for multilingual KGs and larger KGs very well with F-Measure values between 96% and 99% in most cases. As before MultiKE with the MLP classifier performs the best

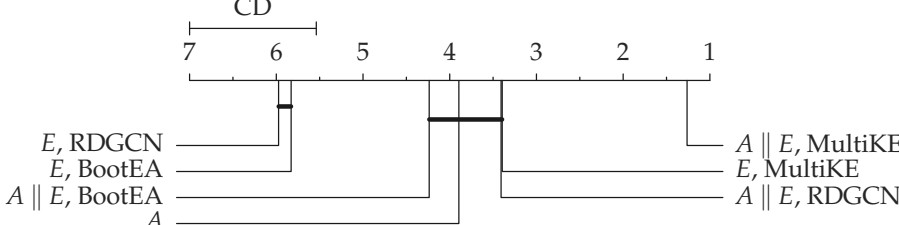

Fig. 4: Critical distance diagram of Nemenyi test, connected groups are not significantly different (at $p = 0.05$)

out of all graph embedding approaches, which is due to the fact that it explicitly takes advantage of attribute information of each entity, as opposed to BootEA and RDGCN.

Comparing the performances between the datasets we see that on the variants with richer graph structure (V2) the results are better than on (V1) for the respective datasets. There is also a difference when contrasting the different sizes of the datasets. While $\text{EAGER}_{A\|E}$ with BootEA and MultiKE generally seem to achieve better results on the larger 100K datasets compared to their 15K counterparts, this is less true for RDGCN.

To properly compare the performance of the approaches across all approaches we used the statistical analysis presented by Demšar [7] and the Python package Autorank [12], which aims to simplify the use of the proposed methods by Demšar. The performance measurement for each dataset and classifier are our paired samples. Given that we have more than two datasets simply using hypothesis tests for all pairs, would result in a multiple testing problem, which means the probability of accidentally reporting a significant difference would be highly increased. We therefore use the procedure recommended by Demšar: First we test if the average ranks of algorithms are significantly different using the Friedman test. If this is the case we perform a Nemenyi test to compare all classifiers and input combinations.

The null hypothesis of the Friedman test can be rejected ($p = 6.572 \times 10^{-25}$). A Nemenyi test is therefore performed and we present the critical distance diagram in Figure 4. The axis shows the average rank of the input/embedding combination. Groups that are connected are not significantly different at the significance level of 0.05, which is internally corrected to ensure that all results together fulfill this. Approaches that have a higher difference in average rank than the critical distance (CD) are significantly different.

We can see that $\text{EAGER}_{A\|E}$ with MultiKE significantly outperforms all other variants. This is evidence that the combination of attribute similarities and embeddings is preferable to using attribute similarities or embeddings on their own for the task of entity resolution in rich knowledge graphs, even if the embedding approaches already incorporate attribute information.

### 5.3 Comparison with other approaches

We compare our approach to the state-of-the-art ER frameworks Magellan [16] and DeepMatcher [19]. Magellan is an ER framework that allows the use of ML classifiers for ER. We present the best performing classifiers XGBoost [5] and Random Forest (RF). DeepMatcher provides several deep learning solutions for ER, we employ the *hybrid* variant which uses a bidirectional recurrent neural network with a decomposable attention-based attribute summarization module. To avoid any decrease in performance due to blocking we provide both frameworks with respective training or test entity mappings directly. Because such a setup is not possible for the approaches discussed in [28], which mostly use resolution strategies based on nearest neighbors, we cannot fairly compare our approach with theirs and therefore refrain from this comparison here.

Table 4: Averaged F-measure, Precision and Recall on test set of rich graph datasets. The best F-measure value in a row is highlighted

| Dataset | EAGER MLP | | | EAGER RF | | | DeepMatcher | | | Magellan XGBoost | | | Magellan RF | | |
|---|---|---|---|---|---|---|---|---|---|---|---|---|---|---|---|
| | fm | prec | rec | fm | prec | rec | fm | prec | rec | fm | prec | rec | fm | prec | rec |
| imdb-tmdb | 0.990 | 0.987 | 0.993 | 0.986 | 0.979 | 0.994 | 0.983 | 0.970 | 0.996 | 0.996 | 0.999 | 0.994 | 0.997 | 0.998 | 0.997 |
| imdb-tvdb | 0.979 | 0.965 | 0.994 | 0.974 | 0.951 | 0.999 | 0.989 | 0.981 | 0.998 | 0.991 | 0.990 | 0.992 | 0.992 | 0.989 | 0.995 |
| tmdb-tvdb | 0.991 | 0.992 | 0.990 | 0.985 | 0.988 | 0.982 | 0.987 | 0.977 | 0.997 | 0.993 | 0.991 | 0.994 | 0.995 | 0.993 | 0.997 |
| D-W(V1) *(15K)* | 0.898 | 0.989 | 0.823 | 0.872 | 0.991 | 0.779 | 0.876 | 0.844 | 0.910 | 0.837 | 0.886 | 0.793 | 0.823 | 0.865 | 0.784 |
| D-W(V2) | 0.968 | 0.990 | 0.948 | 0.909 | 0.992 | 0.838 | 0.904 | 0.895 | 0.913 | 0.863 | 0.899 | 0.830 | 0.848 | 0.859 | 0.837 |
| D-Y(V1) | 0.985 | 1.000 | 0.971 | 0.985 | 1.000 | 0.971 | 0.979 | 0.974 | 0.983 | 0.971 | 0.985 | 0.957 | 0.971 | 0.984 | 0.958 |
| D-Y(V2) | 0.996 | 0.999 | 0.993 | 0.993 | 0.999 | 0.986 | 0.986 | 0.985 | 0.987 | 0.974 | 0.972 | 0.977 | 0.974 | 0.973 | 0.976 |
| EN-DE(V1) | 0.985 | 0.996 | 0.973 | 0.984 | 0.995 | 0.973 | 0.971 | 0.976 | 0.966 | 0.969 | 0.992 | 0.948 | 0.962 | 0.977 | 0.948 |
| EN-DE(V2) | 0.992 | 0.996 | 0.988 | 0.989 | 0.997 | 0.982 | 0.974 | 0.967 | 0.982 | 0.973 | 0.993 | 0.954 | 0.969 | 0.984 | 0.955 |
| EN-FR(V1) | 0.980 | 0.995 | 0.965 | 0.973 | 0.994 | 0.952 | 0.956 | 0.959 | 0.953 | 0.953 | 0.983 | 0.924 | 0.952 | 0.979 | 0.926 |
| EN-FR(V2) | 0.990 | 0.998 | 0.982 | 0.990 | 0.996 | 0.984 | 0.966 | 0.963 | 0.970 | 0.971 | 0.992 | 0.951 | 0.970 | 0.992 | 0.950 |
| D-W(V1) *(100K)* | 0.873 | 0.996 | 0.777 | 0.864 | 0.990 | 0.767 | 0.926 | 0.907 | 0.945 | 0.815 | 0.907 | 0.741 | 0.812 | 0.896 | 0.742 |
| D-W(V2) | 0.965 | 0.989 | 0.941 | 0.926 | 0.988 | 0.871 | 0.936 | 0.924 | 0.949 | 0.836 | 0.925 | 0.762 | 0.831 | 0.897 | 0.774 |
| D-Y(V1) | 0.991 | 1.000 | 0.982 | 0.988 | 1.000 | 0.977 | 0.992 | 0.990 | 0.994 | 0.984 | 0.994 | 0.974 | 0.983 | 0.991 | 0.974 |
| D-Y(V2) | 0.997 | 0.999 | 0.995 | 0.991 | 1.000 | 0.982 | 0.993 | 0.993 | 0.994 | 0.985 | 0.983 | 0.987 | 0.984 | 0.982 | 0.987 |
| EN-DE(V1) | 0.990 | 0.997 | 0.982 | 0.982 | 0.997 | 0.968 | 0.972 | 0.971 | 0.972 | 0.968 | 0.990 | 0.946 | 0.967 | 0.988 | 0.946 |
| EN-DE(V2) | 0.993 | 0.997 | 0.990 | 0.987 | 0.997 | 0.978 | 0.975 | 0.972 | 0.978 | 0.968 | 0.993 | 0.945 | 0.966 | 0.987 | 0.946 |
| EN-FR(V1) | 0.980 | 0.997 | 0.964 | 0.969 | 0.994 | 0.944 | 0.956 | 0.959 | 0.953 | 0.947 | 0.988 | 0.910 | 0.946 | 0.985 | 0.911 |
| EN-FR(V2) | 0.989 | 0.995 | 0.983 | 0.981 | 0.992 | 0.970 | 0.964 | 0.959 | 0.969 | 0.964 | 0.991 | 0.938 | 0.962 | 0.987 | 0.938 |
| Avg Rank | 1.579 | | | 2.579 | | | 2.895 | | | 3.737 | | | 4.211 | | |

We start with the comparison for the shallow datasets. Since both Magellan and DeepMatcher expect matched schemata we align the attributes by hand where necessary. We report F-measure (fm), Precison (prec) and Recall (rec) averaged over the 5 folds. For the comparison with other approaches we use $EAGER_{A\|E}$ with MultiKE and for brevity we will refer to it simply as EAGER. The results are shown in Table 4. All frameworks perform very well with almost all F-measure values over 0.95.

For all three movie datasets Magellan RF outperforms all other approaches in terms of F-measure.

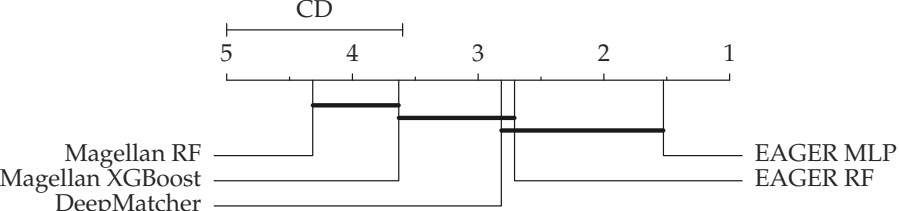

Fig. 5: Critical distance diagram of Nemenyi test for comparison of frameworks, connected groups are not significantly different (at $p = 0.05$)

For the rich graph datasets the heterogeneity of the different KGs was a problem for Magellan and DeepMatcher since they both expect perfectly matched schemata. This was manageable for the smaller datasets, were this can be done by hand. In order to use Magellan and Deepmatcher on the rich graph datasets we did the same as for EAGER and concatenated all entity attributes into a single attribute. We can see that EAGER using MLP outperforms all other approaches except on D-W (V1) and D-Y (V1) for the 100K sizes, where DeepMatcher performs best. Magellan is outperformed on all datasets by EAGER and DeepMatcher.

The Friedman test shows a significant difference ($p = 8.675 \times 10^{-07}$). Looking at the critical distance diagram in Figure 5 we can see that EAGER MLP does not significantly outperform EAGER RF or DeepMatcher, but it is the only approach that significantly outperforms both Magellan approaches. While there is no significant difference between EAGER MLP and DeepMatcher, EAGER does not depend on the provision of schema matching.

## 6    Conclusion & Future work

We explored the combination of knowledge graph embeddings and attribute similarities for entity resolution in knowledge graphs with multiple entity types. These approaches are included in a new learning-based ER system called EAGER. We tested our approach on a range of different datasets and showed that using a combination of both graph embeddings and attribute similarities generally yields the best results compared to just using either one. We showed that our approach yields competitive results that are on par with or significantly outperform state of the art approaches. The approach is generic and can deal with several entity types without prior schema matching.

Future work will investigate blocking strategies utilizing both embeddings and attribute information, as well as smarter attribute combination strategies (e.g. using property matching[2]). Unsupervised and active learning in this context should be explored to alleviate the difficulty of obtaining training data.

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
