# OpenReview forum: "Embedding-Assisted Entity Resolution for Knowledge Graphs"
_eswc-conferences.org/ESWC/2021/Workshop/KGCW — KGCW 2021_

### Official Review · ~Femke_Ongenae1 · 2021-04-10
**Clear paper that nicely evaluates the benefit of combining two prominent approaches (graph embedding approaches & attribute-based ones) for entity resolution within Knowledge Graphs**

**Rating:** 8
**Confidence:** 3

**Review:**

This paper presents a new approach, called EAGER, for performing entity resolution between Knowledge Graphs (KGs), by combining prevalent methods from two existing approaches, i.e. graph embedding based approaches & attribute-based ones.

The paper is written very clearly and is easy to follow. It gives a comprehensive overview of the related work within the field & how EAGER fits within it. While the methodology to combine both approaches is rather straightforward (concatenation of the vectors created by each), the real contribution of the paper, in my opinion, lies in the thorough and excellent evaluation. The evaluation methodology is clearly explained and is solid. Moreover, the paper contains an in depth study and discussion of the different results. As such, one gets a clear view for which cases the combined approach of EAGER performs better (or similar/worse) than the individual approaches and what the underlying reason is for this performance gain (or loss). As such, this paper lays important groundwork for future investigation towards combined approaches for entity resolution, and the added value of using graph embeddings within this field. I also commend the authors for making the code available for the presented method, as well as the performed experiments and created data sets. Again, this contributes towards further enhancements in this field.

Some minor comments:
- The quotation marks around words are not correctly formatted
- On the first page (entity, property, value) goes beyond the page borders
- Some sentences are rather long (spanning for more than three lines), without commas, making them hard to digest. I would suggest splitting them up into smaller, easier to digest sentences. (An example is the second last sentence of Section 2)
- The second sentence of Section 2 has an incorrect "." in the middle that should be a comma "," (right before the "and")

---

### Official Review · ~Heiko_Paulheim1 · 2021-04-12
**System which yields good results, but some aspects in the approach and evaluation protocol need more discussion**

**Rating:** 6
**Confidence:** 4

**Review:**

The paper introduces EAGER, an entity alignment tool for knowledge graphs which combines attribute value similarities with entity embeddings.

The approach uses a two-part feature vector for representing a pair of entities: the first part is based on similarity of the (concatenated) attribute values, the second part is comprised of the entity embedding vectors of the two entities. A downstream classifier learns to distinguish true and false matches.

On the pro side, the evaluation is very well conducted, uses a variety of datasets, compares against relevant state of the art approaches, and uses statistical significance testing. However, since there has also been a shared task at OAEI for a few years now [1], I am curious to hear why the authors did not evaluate against that shared task as well (which would have yielded a direct comparison against a few more competitive systems).

Although the evaluation setup is following another paper published at VLDB, I still want to raise some doubts about the evaluation protocol. Using arbitrary pairs of entities as negative examples seems to overly simplify the task. For an arbitrary pair of entities, for example, I expect the similarity of the concatenated attribute values to be close to 0 anyways, so telling a true match from an arbitrary pair of entities seems a not too difficult task (it would be nice to show distributions of the attribute similarities both for the positive and negative examples to prove me wrong here).

Knowledge graph matching, however, is different from telling matches from arbitrary pairs of entities. The hard challenges in knowledge graph matching are the cases of related, but different entities (e.g., a Batman movie and the fictional Batman character), or pairs of entities that have similar names (and maybe even types), but are yet dissimilar (e.g., different Batman movies). Such corner cases are unlikely to appear in the random negative examples, thus, it is unclear how the approach would behave in the wild. For evaluation, a more realistic estimate of precision could be obtained by using the protocol also used for the OAEI KG track, i.e., assuming that there is only one correspondent entity in KG2 for each entity in KG1 (and vice versa), and thus consider (E1xE2)\M as a set of negative examples.

Speaking of application in the wild: the authors acknowledge themselves that they can, so far, only test given pairs of entities, but not produce a full alignment between the two knowledge graphs, because the approach does not come with a blocking strategy. Here, it would be interesting to see the approach at work with a simple blocking strategy to prove that it actually works on large scale knowledge graphs.

One aspect of the approach that left me a bit puzzled was the usage of attribute similarities. While I can see that this might work for a few quite text heavy literals, literals may also be numbers or dates. Here, it seems like the usage of Levenshtein or trigram similarity on concatenated literals is unlikely to yield helpful information. Maybe they do not appear too frequently in the test datasets, but in my humble opinion, this aspect should be discussed a bit more thoroughly.

Finally, while the approach is a generic setup which can combine any embedding method with attribute similarities, I wonder if the same result could also be achieved by directly incorporating literals into the embedding [2].

Summarizing: some aspects of the approach and the evaluation setup do not fully convince me. Still, there might be interesting discussions if this work is presented at the workshop.

[1] http://oaei.ontologymatching.org/2020/knowledgegraph/index.html
[2] http://www.semantic-web-journal.net/content/survey-knowledge-graph-embeddings-literals-which-model-links-better-literal-ly-0

---

### Official Review · ~Giuseppe_Futia1 · 2021-04-14
**Interesting tool and promising results**

**Rating:** 7
**Confidence:** 3

**Review:**

The paper presents a new system called EAGER, which combines graph embeddings and attribute values to address the Entity Resolution (ER) task. The article is well structured, the method adopted by the authors is technically compelling, and the evaluation procedure is clear and well-conducted. I appreciated the distinction between rich and shallow graphs in the evaluation process: it could represent a useful proxy for guiding the EAGER user in selecting the appropriate method for his specific use case or dataset.

Although the authors have selected 3 of the best approaches included in OpenEA, I believe that a further discussion on the role of graph embeddings for the EA task would complement the contributions of the article. From a theoretical perspective, the graph (or network) embedding problem is related to different signals that can be derived and learned from graph data. For instance, we can distinguish between positional and structural embeddings: techniques such as matrix factorization preserve the global structure, reconstructing the edges in the graph, and maintaining distances such as the shortest paths in the original network. On the other side, Graph Neural Networks, such as GCN, intend to capture the local graph structure: nodes with similar neighbors in a graph should have similar embeddings. For future work, I suggest the authors study the impact of these types of differences in the ER downside task.

To summarize, I believe that the contribution of the authors is relevant and that they have identified a promising approach.

---

### Meta-Review · Program_Chairs · 2021-04-21

**Recommendation:** Accept
**Confidence:** 5

**Metareview:**

All reviewers agree that the paper is written very clearly and is easy to follow. It gives a comprehensive overview of the related work within the field, a thorough and well-conducted evaluation and in depth study and discussion of the different results. The paper uses a variety of datasets, compares against relevant state of the art approaches, and uses statistical significance testing. The paper lays important groundwork for future investigation and we would like to compliment the authors together with the reviewers for making the code available for the presented method, as well as the performed experiments and created data sets.

The reviewers also suggested some more topics that would be interesting to be discussed in the paper, such as the role of graph embeddings for the EA task, the usage of attribute similarities and why EAGER was not evaluated against the OAEI shared task yielding a direct comparison against more competitive systems. Then a reviewer expressed some opportunities for discussion and further improvements of the work regarding the evaluation protocol, precision estimation using the protocol used for the OAEI KG track, a simple blocking strategy for large scale knowledge graphs, directly incorporating literals into the embedding etc. These are interesting topics to spark fruitful discussions during the workshop.

---

### Decision · Program_Chairs · 2021-04-23

Accept